# Closing the Gap between the Auditory Nerve and Cochlear Implant Electrodes: Which Neurotrophin Cocktail Performs Best for Axonal Outgrowth and Is Electrical Stimulation Beneficial?

**DOI:** 10.3390/ijms24032013

**Published:** 2023-01-19

**Authors:** Dominik Schmidbauer, Stefan Fink, Francis Rousset, Hubert Löwenheim, Pascal Senn, Rudolf Glueckert

**Affiliations:** 1Inner Ear Laboratory, Department of Otolaryngology, Medical University of Innsbruck, 6020 Innsbruck, Austria; 2Translational Hearing Research, Tübingen Hearing Research Center, University of Tübingen, 72076 Tübingen, Germany; 3The Inner Ear and Olfaction Laboratory, Department of Pathology and Immunology, Faculty of Medicine, University of Geneva, 1206 Geneva, Switzerland; 4Department of Otolaryngology, Head and Neck Surgery, University of Tübingen, 72076 Tübingen, Germany; 5Department of Clinical Neurosciences, Service of ORL & Head and Neck Surgery, University Hospital of Geneva, 1205 Geneva, Switzerland; 6University Clinic for Ear, Nose and Throat Diseases, Tirol Kliniken, 6020 Innsbruck, Austria

**Keywords:** nerve regeneration, cochlear implant, spiral ganglion, neurotrophin treatment, electrical stimulation

## Abstract

Neurotrophins promote neurite outgrowth of auditory neurons and may help closing the gap to cochlear implant (CI) electrodes to enhance electrical hearing. The best concentrations and mix of neurotrophins for this nerve regrowth are unknown. Whether electrical stimulation (ES) during outgrowth is beneficial or may direct axons is another open question. Auditory neuron explant cultures of distinct cochlear turns of 6–7 days old mice were cultured for four days. We tested different concentrations and combinations of BDNF and NT-3 and quantified the numbers and lengths of neurites with an advanced automated analysis. A custom-made 24-well electrical stimulator based on two bulk CIs served to test different ES strategies. Quantification of receptors trkB, trkC, p75^NTR^, and histological analysis helped to analyze effects. We found 25 ng/mL BDNF to perform best, especially in basal neurons, a negative influence of NT-3 in combined BDNF/NT-3 scenarios, and tonotopic changes in trk and p75^NTR^ receptor stainings. ES largely impeded neurite outgrowth and glia ensheathment in an amplitude-dependent way. Apical neurons showed slight benefits in neurite numbers and length with ES at 10 and 500 µA. We recommend BDNF as a potent drug to enhance the man-machine interface, but CIs should be better activated after nerve regrowth.

## 1. Introduction

Approximately one in five people worldwide experience some form of hearing impairment and about 49.5 million people suffer from severe to complete hearing loss [1]. Absolute numbers of these patients are expected to rise as life expectancy increases, and hearing loss is closely related to aging. Possible consequences of hearing loss include social isolation and loneliness [2], increased risk of dementia [3], and cognitive decline in elderly people [4]. If hearing aids are not able to restore hearing sufficiently, cochlear implants (CI) are the state-of-the-art therapy. The main cause for sensorineural hearing loss is damage of hair cells within the organ of Corti. CIs aim at replacing hair cell functionality by delivering electrical stimulation (ES) via an electrode array inserted into the cochlea. The stimulation targets are the spiral ganglion neurons (SGNs),bipolar neurons innervating the sensory hair cells. The somata of the SGNs are enclosed in Rosenthal’s canal, a helical tunnel in the modiolus. As conventional CI electrode arrays are mostly situated along the lateral wall of the scala tympani or close to the modiolar wall after insertion, a spatial gap between the electrodes and their stimulation targets exists [5]. Closing this anatomical gap could lead to significantly improved stimulation outcomes because of less current spread and consecutively less electrode crosstalk as well as reduced power consumption [6,7]. One proposed solution to diminish this gap is the regrowth of peripheral processes towards the electrode array [8].

Both brain-derived neurotrophic factor (BDNF) and neurotrophin-3 (NT-3) are known as potent neurotrophins (NTs) to promote SGN survival and even to induce resprouting and regrowth of neurites in single-cell cultures [9,10,11,12,13,14,15], organotypic explant cultures [11,14,16,17,18,19,20,21] and in vivo [22,23,24,25,26,27,28,29,30,31,32,33,34,35,36]. BDNF binds with high affinity to the tyrosine receptor kinase B (trkB), while NT-3 does so to the tyrosine receptor kinase C (trkC). The effects of trk signaling include promotion of neuronal survival and the stimulation of neurite growth [37]. Both NTs bind to the p75 neurotrophin receptor (p75^NTR^) with lower affinity. This interaction leads to atrophic effects, such as neurite growth inhibition and promotion of apoptosis [38]. BDNF, NT-3, and their receptors also play an important role during development for SGN differentiation and hair cell innervation [39,40] through spatiotemporal expression patterns along the tonotopic axis [41,42]. To our knowledge, a systematic investigation of dose-dependent effects of these two NTs on outgrowth characteristics of SGN explants at different tonotopic locations has not been performed yet. 

ES is another promising though controversially discussed treatment to enhance SGN survival after hair cell loss. Previous studies found that ES can preserve SGN density compared to the untreated contralateral side in vivo [30,43,44,45,46,47,48,49,50]. Other studies, however, did not observe a beneficial outcome following ES in vivo [51,52,53,54,55,56]. ES applied to SGNs cultured in vitro gives a clearer picture: SGN survival [57] and neurite extension [58] were diminished by ES in dissociated SGN cultures. Moreover, ES decreased the number of peripheral processes [59] and induced neurodegeneration [60] in organotypic explant cultures. Beyond that, ES may also be used to direct regrowing neurites toward the tonotopically corresponding electrode contacts. Closing the spatial gap between the electrodes and the peripheral processes may not improve stimulation selectivity if neurites regrow in a disorganized way and spread randomly into the scala tympani. To shed more light on the question whether ES is beneficial at the time of neurite regrowth and whether it could support guidance, we developed a novel electrode inset for a medium-throughput experimental approach. Subsequently, we tested ascending amplitudes but also an interrupted pattern to simulate the spontaneous firing pattern of immature inner hair cells at this point in development [61].

In in vivo experiments, neurite outgrowth is relatively difficult to quantify because regrowing fibers penetrate different compartments of the cochlea, grow three-dimensionally, and often fasciculate in bundles [36]. Sophisticated imaging techniques or time-consuming sectioning are needed to assess the entire outgrowth. Consequently, sample numbers are rather low in such an approach, limiting the number of treatment conditions to be tested. As our goal was to quantify resprouting fibers, we chose to culture organotypic explants from the spiral ganglion. This primary tissue provides an in vivo-like cellular environment [62], containing Schwann cells and satellite glia. These cells express p75^NTR^, the low-affinity receptor for BDNF and NT-3, which is upregulated after axotomy [63]. They are known to play an important role in the complex neurotrophic signaling pathways. Moreover, outgrown fibers can be quantified more easily as they grow in an unobstructed and flat way on a coated cell culture dish.

## 2. Results

In this study, we investigate the acute effects of BDNF, NT-3, combinations thereof, and ES on neurite regrowth of organotypic explants along the tonotopic axis in a medium throughput setup. Apical, middle, and basal SGN explants were cultured for 4 days with different NT concentrations added. The best-performing neurotrophin cocktail was further used for ES to identify whether ES can improve elongation and direct neurite outgrowth or should not be used immediately after NT administration. After fixation, neurites (number and length) and p75^NTR^/trk-receptor immunostainings were quantified. Plastic-embedded sections exposed the detailed morphology of each treatment group. Additionally, we shed more light on organotypic explant cultures of the spiral ganglion as a model for neuronal regeneration in the cochlea by analyzing correlations of categorical measures such as sex, body side and age, and numerical factors such as explant area, branching points or body weight on neurite outgrowth.

All values are given as the median of the treatment groups, as most data were not normally distributed. The results of the NT and ES experiments are denoted as fold changes relative to the median of the respective untreated control group. This emphasized the actual effect of NT and ES treatment and was considered to be more relevant for future applications than absolute measurements. Sample sizes for each group are denoted below the x-axis in the respective figures. 

### 2.1. Factors of Influence on Neurite Outgrowth in Explant Cultures

In total, 663 explants (*n* = 663) were used for the NT study and 427 (*n* = 427) for the ES study. Two large datasets were available to analyze fundamental correlations between different variables: The untreated control group of the NT experiments (NT CTRL, *n* = 72) and the unstimulated but BDNF supplemented (25 ng/mL, best treatment for outgrowth) control group of the ES experiments (ES CTRL, *n* = 213). Two different types of statistical variables may be distinguished: continuous variables can attain any numeric value within a certain range (e.g., explant area), while categorical variables describe mostly non-numerical categories (e.g., body side). Age was considered a categorical variable because only two ages (P6 and P7) were utilized here.

The influence of the categorical measures, such as sex, body side, and age, was investigated using groupwise comparisons. In both datasets, the sex of the mice did neither influence the numbers nor the median lengths of the outgrowing neurites significantly, but explants from females produced slightly more neurites on average (NT CTRL: 117 vs. 78, *p* = 0.4366; ES CTRL: 420 vs. 352, *p* = 0.2184). The same was true for the body side where the explants were extracted from. Explants from the right side were always prepared first; hence the idle time of ~10 min in ice-cold HBSS of the left ears did not affect the outgrowth performance significantly. Although both numbers of neurite endings (NT CTRL: 117 vs. 104, *p* = 0.9931; ES CTRL: 416 vs. 356, *p* = 0.3120), as well as median lengths (NT CTRL: 340 µm vs. 326 µm, *p* = 0.4247; ES CTRL: 564 µm vs. 506 µm, *p* = 0.0735), were slightly higher in explants from the right side. The age of the mice -P6 or P7- did also not influence the number of neurite endings and the median lengths significantly, but neurites from P6 explants tended to grow longer (NT CTRL: 390 µm vs. 326 µm, *p* = 0.6124; ES CTRL: 555 µm vs. 518 µm, *p* = 0.1329).

Continuous variables were examined by means of correlation analysis (Appendix A). Mouse body weight and number of neurite endings, or median neurite lengths, respectively, did not show significant correlations. A negative correlation between mouse body weight and the projected area of the explants was found (NT CTRL: −0.25, *p* = 0.0356; ES CTRL −0.13, *p* = 0.0676). The explant area exhibited slightly positive correlations with all other outgrowth variables, meaning that larger explants may have contained more surviving neurons. This correlation was stronger in the BDNF-supplemented dataset. However, the explant area was subjected to substantial changes during the first days in culture, as observed by live cell imaging (Appendix A). In general, outgrowth-related measures correlated well among themselves. Particularly, the median neurite length of an explant exhibited a rather high correlation coefficient with the number of neurites (NT CTRL: 0.70, *p* < 0.0001; ES CTRL: 0.81, *p* < 0.0001) and the length of the longest neurite (NT CTRL: 0.74, *p* < 0.0001; ES CTRL: 0.67, *p* < 0.0001). The number of neurite endings, the number of branch points, and the total length of the outgrown neurites were all almost perfectly correlated (0.95 < r < 0.98), making them highly tantamount variables. While there was a high correlation between the number of start-points (fiber fascicles exiting the explant body) and the number of neurites (NT CTRL: 0.83, *p* < 0.0001; ES CTRL: 0.72, *p* < 0.0001), the impact on the median neurite lengths was weaker (NT CTRL: 0.40, *p* = 0.0001; ES CTRL: 0.48, *p* < 0.0001).

We categorized the extracted tissue in three distinct locations: the apical, middle, and basal turns. Without neurotrophic supplementation, the median of neurite endings was 107 in explants extracted from the apical turn, 165 from the middle turn, and 59 from the basal turn. The medians of the neurite lengths were as follows: apical turn 316 µm, middle turn 392 µm, and basal turn 311 µm. These values served as references for the NT-supplemented experiments. 25 ng/mL BDNF increased the median number of neurites drastically: 446 in the apical turn, 385 in the middle turn, and 335 in the basal turn. The median lengths were 576 µm (apical turn), 528 µm (middle turn), and 489 µm (basal turn).

In summary, we could show that neither sex nor age or body side influenced outgrowth significantly. Interestingly, explants from mice with higher bodyweight were smaller. The explant area had a slight and expectable positive effect on outgrowth, and most of these outgrowth parameters correlated highly with each other. These findings were evident in untreated as well as BDNF-supplemented explants.

### 2.2. Effects of Neurotrophins on Neurite Outgrowth

Organotypic explants of the spiral ganglion grew with different concentrations of BDNF, NT-3, and combinations thereof in the culture medium. The two potentially most relevant variables for closing the anatomical gap were evaluated in depth: the number of neurite endings (neuron survival and neurite sprouting) and the median of all neurite lengths (neurite elongation).

#### 2.2.1. Number of Neurite Endings

BDNF and NT-3 are both very potent NTs to increase the number of outgrowing neurites (Figure 1). Figure 1A shows the results for the whole cochlea, consisting of equally sampled explants from the three turns. Even at concentrations as low as 1 ng/mL, BDNF increased the median number of neurite endings by a factor of 4.93 (*p* < 0.0001). The potency of NT-3 was less at low concentrations (3.65× NT CTRL^whole cochlea^ at 1 ng/mL, *p* < 0.0001). Despite an unsteady progression, a dose-response dependency of both NTs was noticeable. The best stimulating NT concentrations found were 25 ng/mL (9.93× NT CTRL^whole cochlea^, *p* < 0.0001) for BDNF and 200 ng/mL for NT-3 (6.95× NT CTRL^whole cochlea^, *p* < 0.0001). Treatment with all concentrations of BDNF and most of NT-3 was significantly better than the untreated control. Combining both NTs did not reveal an additive or synergistic effect (6.25× NT CTRL^whole cochlea^ at 200 ng/mL BDNF + 25 ng/mL NT-3, *p* < 0.0001). Examining discrete tonotopic regions, similar patterns in the apical (Figure 1B) and middle turn (Figure 1C) were found. The basal turn (Figure 1D) was much more susceptible to neurotrophic treatment than other turns, especially with BDNF. Explants with an outgrowth closest to the median of the best-performing whole-cochlea groups are depicted in Figure 1E and illustrate their high efficacy on neurite sprouting.

To assess the general potency of a NT or the combinations, the samples of all concentrations were pooled in equal shares and compared to the NT control group. BDNF (5.60× NT CTRL^whole cochlea^, *n* = 162), NT-3 (4.32× NT CTRL^whole cochlea^, *n* = 162), and BDNF + NT-3 (5.41× NT CTRL^whole cochlea^, *n* = 72) all performed significantly better than the non-supplemented controls in the whole cochlea, but also in the single turns. Moreover, BDNF efficacy surpassed the one of NT-3 in the whole cochlea (*p* = 0.0239).

The same pooling can also be used to compare distinct turns with each other. Compared with the apical (3.91× NT CTRL^apical^, *p* = 0.0069, *n* = 72) and middle turn (4.00× NT CTRL^middle^, *p* = 0.0001, *n* = 90), explants from the basal turn (7.31× NT CTRL^basal^, *n* = 54) showed a significant increase in neurite numbers when treated with BDNF. NT-3 supplementation resulted in a significant change in sprouting when comparing the basal turn (5.14× NT CTRL^basal^, *n* = 54) with the middle (3.39× NT CTRL^middle^, *p* = 0.0171, *n* = 72) but not with the apical turn (3.71× NT CTRL^apical^, *p* = 0.1679, *n* = 63). Combined treatment with BDNF and NT-3 did not reveal any significant differences in the apical (4.47× NT CTRL^apical^, *n* = 24), middle (3.71× NT CTRL^middle^, *n* = 24), and basal turns (4.61× NT CTRL^basal^, *n* = 24).

#### 2.2.2. Median Neurite Length

Neurite lengths were considered separately from the number of neurite endings. Each data point is the median of the lengths of all neurites from one explant. Contrary to the number of neurite endings, BDNF and NT-3 did not have such a substantial impact on the neurite lengths (Figure 2). The best BDNF concentration was again 25 ng/mL with a 1.98-fold length increase compared to the untreated control (*p* = 0.0001). NT-3 at a concentration of 200 ng/mL was performing best (1.50× NT CTRL^whole cochlea^, *p* > 0.9999). The best-combined treatment of 200 ng/mL BDNF + 200 ng/mL NT-3 had only a minor but non-significant effect (1.22× NT CTRL^whole cochlea^, *p* > 0.9999). Only some BDNF concentrations resulted in significantly longer neurites, while none of the NT-3 or BDNF + NT-3 treatments did alike. Overall, the middle (Figure 2C) and basal turns (Figure 2D) reacted similarly. Only the NT-3 treatment in the apical turn (Figure 2B) seemed to be more effective. Explants closest to the median of the best-performing groups in Figure 2E exemplified these results on neurite lengths visually.

Explants of all concentrations were again pooled equally (for sample sizes, see Section 2.2.1). Across the whole cochlea, BDNF, NT-3, and the combined treatment (all *p* < 0.0001) all resulted in significantly longer neurites compared to no NT supplementation. Moreover, BDNF treatment (1.75× NT CTRL^whole cochlea^) outperformed both the NT-3 (1.23× NT CTRL^whole cochlea^, *p* < 0.0001) and BDNF + NT-3 treatment (1.13× NT CTRL^whole cochlea^, *p* < 0.0001), while no significant difference was found between NT-3 and BDNF + NT-3 (*p* = 0.8313). The lengths did not differ between the three turns within each of the NT or combination groups, except that the apical turn (1.28× NT CTRL^apical^) responded significantly better to NT-3 than the basal turn (1.09× NT CTRL^basal^, *p* = 0.0443). 

### 2.3. Effects of Electrical Stimulation on Neurite Outgrowth

To ensure that the stimulation inset did not impair culture conditions, fluid evaporation of a filled plate (400 µL/well) with a normal lid was compared with a filled plate with the stimulation inset. The weight of the plate with the normal lid was reduced by 0.40 g, and the one with the stimulation inset was reduced by 0.52 g after 72 h of incubation. Therefore, 0.12 g more liquid evaporated with the simulation inset, which equals to only 5 µg or 1.25% more evaporation per well. Nevertheless, a comparison of the unstimulated controls of the ES experiment (ES CTRL^whole cochlea^, Median: 385, *n* = 201) with the equally treated group of the NT experiment (NT 25 ng/mL, Median: 864, *n* = 24, see Figure 2A)—both supplemented with 25 ng/mL BDNF and randomly equalized regarding their tonotopic location—revealed a significant decrease (Mann–Whitney test, *p* = 0.0005) in neurite numbers. The temperature during stimulation with an amplitude of 1000 µA was measured using miniature temperature sensors placed where the explants would be. No change in temperature (data not shown) was detected in the electrically stimulated wells. 

The explants were electrically stimulated for 72 h after a settling period of 24 h. The amplitude of the biphasic pulse varied while all other stimulation parameters remained unchanged. As considerable variation was evident in the NT experiments, we modified the study design for the ES experiments. Half of the organotypic explants in each plate were not stimulated electrically and served as direct “in-plate” controls. The outgrowth data of the electrically stimulated explants were then normalized to the controls in the same plate to reduce variation. 

#### 2.3.1. Number of Neurite Endings

ES during the outgrowth phase reduced the number of neurite endings in most of the cases (Figure 3). The reduction in median neurite numbers at an amplitude of only 10 µA was already considerable (0.62× ES CTRL^whole cochlea^, *p* = 0.1814) in the whole cochlea (Figure 3A). Higher amplitudes reduced the numbers even further (1000 µA, 0.51× ES CTRL^whole cochlea^, *p* = 0.0069). The one-minute-on-two-minute-off pattern at 1000 µA led to a similar result (0.49× ES CTRL^whole cochlea^, *p* = 0.0003). Apical turn explants were more robust against ES (Figure 3B), as the median of the stimulated groups was close to the ones of the control groups without a statistically significant difference. Only the on-off pattern reduced the number of neurites significantly (0.49× ES CTRL^apical^, *p* = 0.0014), while 500 µA even increased outgrowth slightly (1.11×, *p* = 0.5668). None of the stimulation patterns decreased neurite numbers significantly in the middle turn (Figure 3C), and amplitudes of 100 and 500 µA impeded neurite sprouting the least. ES considerably affected explants from the basal turn (Figure 3D), but only 100 µA did so significantly (0.30× ES CTRL^basal^, *p* = 0.0075).

All different stimulation groups were again pooled in equally sized cohorts for statistical comparison. In the whole cochlea, ES resulted in a 0.59 (*p* < 0.0001, *n* = 104) fold change in neurite numbers compared to electrically unstimulated controls (ES CTRL^whole cochlea^, *n* = 135). Explants from the apical turn (ES^apical^: *n* = 45, ES CTRL^apical^: *n* = 50) did not produce significantly fewer neurites (0.81× ES^apical^, *p* > 0.9999), while they did so when extracted from the middle (0.59× ES^middle^, *p* = 0.0116, ES^middle^: *n* = 45, ES CTRL^middle^: *n* = 70) or basal turns (0.41× ES CTRL^basal^, *p* = 0.0027, ES^basal^: *n* = 35, ES CTRL^basal^: *n* = 45). Comparing the turns with each other revealed that neurite outgrowth performance in the basal turn was significantly worse than in the apical turn (*p* = 0.0120). The poor outgrowth in the basal turn with ES is in stark contrast to BDNF treatment without ES.

#### 2.3.2. Median Neurite Length

The impact of ES on the median neurite lengths was not as strong as on the neurite numbers (Figure 4). The inhibitory effect of the stimulation was amplitude-dependent in the whole cochlea (Figure 4A). Neurites were shorter with increasing current levels, resulting in significantly shorter lengths at 1000 µA (0.74× ES CTRL^whole cochlea^, *p* = 0.0016). Reducing the total stimulation time at 1000 µA by the on-off pattern diminished neurite length less (0.84× ES CTRL^whole cochlea^, *p* = 0.0168) than the continuous stimulation. Neurite lengths of explants from the apical turn were less affected (Figure 4B). Mild stimulation with 10 µA even improved neurite lengths by a factor of 1.19 (*p* = 0.3471). On the contrary, explants from the middle (Figure 4C) and basal turns (Figure 4D) exhibited comparatively shorter neurites.

Pooling all stimulation patterns (for sample sizes, see 3.3.1) revealed that lengths were reduced by a factor of 0.83 in the whole cochlea (*p* < 0.0001). The lengths in the apical turn were not significantly diminished (0.87× ES CTRL^apical^, *p* = 0.3959) in contrast to the middle (0.89× ES CTRL^middle^, *p* = 0.0440) and the basal turns (0.79× ES CTRL^basal^, *p* = 0.0016). There was no significant difference in neurite lengths amongst the turns.

#### 2.3.3. Outgrowth Direction

To estimate whether ES during the outgrowth phase has an influence on the growth direction, the angle between the vertical image axis and a vector, connecting the point where a neurite supposedly exits the explant body and the tip of this neurite, was measured. As the electrical field was always orientated in the same direction regarding the vertical image axis, a significant change in the fibers’ orientation should be detectable. However, no such deviations were observed comparing the unstimulated with the stimulated explants (Figure 5).

### 2.4. Quantification of Neurotrophic Receptors

The expression of three neurotrophic receptors—trkB, trkC, and p75^NTR^—was measured in a semi-quantitative manner to compare treatment groups. Sections of immediately fixed inner ears at different ages (P0, P7 and P11) and explants with selected treatments (untreated (NT CTRL), 25 ng/mL BDNF, 200 ng/mL NT-3, 25 ng/mL BDNF + 200 ng/mL NT-3 and 25 ng/mL BDNF + 500 µA) were processed immunohistochemically, and the mean intensity of DAB-positive areas was computed. All intensity values in the following section are given as the median immunoreactivity (IR) of all explants within the group, as data distribution was not normal.

#### 2.4.1. TrkB

Across the whole cochlea, an age-dependent decline (P0: 214, P7: 101, P11: 64) of trkB IR in the postnatal inner ears was evident (Figure 6A). IR is a dimensionless unit ranging from 0 (no staining) to 255 (saturation). The untreated control explants exhibited significantly higher levels of trkB receptor IR (139, *p* = 0.0234) than P11 mice. This age was chosen because it corresponds to the same timespan of postnatal development as the explants lived through since birth. Treatment with BDNF reduced the expression (105) to approximately the same level as in P7 mice, while NT-3 supplementation barely affected the expression levels (143) compared to the controls. A combination of both factors reduced trkB receptor IR (88) slightly more than BDNF alone. ES + BDNF did not influence these levels (106) compared to BDNF alone when considering the whole cochlea.

Distinct cochlear turns expose the same pattern. However, trkB IR was considerably reduced in the electrically stimulated apical explants (Figure 6B), while the middle-turn IR intensities (Figure 6C) resembled the ones of the whole cochlea. In the basal turn (Figure 6D), the decrease of IR intensity followed by BDNF and BDNF + NT-3 supplementation was most prominent. Remarkably, ES eliminates the BDNF-mediated decrease in IR intensity in the basal turn. We observed a gradient of IR along the tonotopic axis from apex to base in P0 (220, 215, 205) and P7 (123, 103, 94) mice.

#### 2.4.2. TrkC

TrkC receptor expression declined from P0 (158) to older stages in the whole cochlea, though the differences between P7 (63.59) and P11 (69) were insignificant (Figure 7A). IR intensity in untreated controls (NT CTRL) was higher (103) than in the corresponding age-matched group at P11. BDNF (79), BDNF + NT-3 (67), and ES (57) led to a decrease in IR compared to the control group, while it was almost unaltered after supplementation with its high-affinity ligand NT-3 (97).

IR intensities in the untreated control group (NT CTRL) were elevated in the middle (Figure 7C) and rather low in the basal turn (Figure 7D) in comparison with the apical turn (Figure 7B). An opposing effect was found after NT-3 treatment. Interestingly, BDNF supplementation led to a considerably reduced trkC expression in the basal turn. BDNF + NT-3 produced higher mean intensities in the middle turn compared to flanking tonotopic regions (apical and basal). ES led to reduced IR intensities in the apical and middle turns compared to BDNF alone, while ES results in higher IR levels in the basal turn. In P7 mice, a slight tonotopical decline in trkC IR from apex to base (66, 64, 63) was observed.

#### 2.4.3. P75^NTR^

Mean IR intensities of the p75^NTR^ receptors also declined with postnatal age. In the whole cochlea (Figure 8A), mean IR intensity decreased from 92 at P0, to 32 at P7 and to 24 at P11. Explantation in general resulted in a massive and statistically significant upregulation of the p75^NTR^ receptor expression compared to P11 acute preparations. Without treatment (NT CTRL), the median of the intensities was 195, followed by 200 ng/mL NT-3 (179) and the electrically stimulated group (171). BDNF supplementation-alone or in combination with NT-3- resulted in further reduced IR intensities 148 and 130 respectively).

IR intensities of the control groups were quite consistent among turns. In explants from the middle turn (Figure 8C), p75^NTR^ IR intensity was higher after BDNF application compared to the apical (Figure 8B) or basal turn (Figure 8D). ES resulted in higher staining in the basal turn when compared to the apical and middle turn. In P0 mice, an apex-to-base IR gradient was evident (100, 92, 84). Moreover, Figure 8F,G demonstrates that p75^NTR^ IR was limited to non-SGN cells, mainly Schwann cells and satellite glia.

### 2.5. Explant Morphology

As the explant shape changed dramatically during the cultivation period (Appendix A) and outgrowth performance differed among treatment groups, we processed representative individual cultures (closest to the median value of the number of outgrown neurites) for semithin sectioning to evaluate morphological differences. Unsupplemented explants (NT CTRL) showed many degenerating neurons, often associated with their satellite glial cells (SGC) (Figure 9A–C). Application of 1 (not shown) to 25 ng/mL BDNF resulted in the most “natural” cellular arrangement with very big SGNs ensheathed by plenty of SGCs. The tissue was pervaded by tracks of parallel fibers associated with nuclei exhibiting a morphology typical for Schwann cells (Figure 9D,E). Prominent nucleoli indicate a high level of DNA transcription. The best outgrowth-performing NT-3 supplementation with 200 ng/mL caused smaller-sized neurons with fewer SGCs around SGNs and a brighter cytoplasm (Figure 9F,G). The best-performing combined NT treatment provoked a similar morphology as BDNF 25 ng/mL alone (Figure 9H,I). ES with 25 ng/mL BDNF at 100 µA diminished the size of the SGNs and led to a reduction of SGCs with very small and intensely stained nuclei, indicating degenerative changes (Figure 9J,K). A further increase in amplitude to 1000 µA with a 1-min-on-2-min-off pattern (the worst performing ES group) intensified this effect in a way that many neurons are no longer associated with their SGCs and the cytoplasm of SGNs appeared pale and full of vacuoles (Figure 9L–N). The trophic effect of BDNF and the damaging impact of ES were more pronounced in SGCs and other (glial) cells, visible by their pyknotic cell nuclei (Figure 9J,L).

### 2.6. Results Summary

The plethora of data requires some compression for a comprehensive view of the main effects of NT treatment and ES. Figure 10 aims to break down the most prominent effects in a color-coded way. BDNF boosted neurite sprouting and likely neuron survival most prominently in the basal turn, leading to the longest neurites. BDNF was the clear winner in sprouting performance. These effects went along with a reduction of trkB and C receptor expression with similar tonotopic gradients. NT-3 had less pronounced effects on neurite amount and length. The effect of NT-3 on trk receptors deviated from the one of BDNF. IR intensities even rose partially, most pronounced in the basal turn. P75^NTR^ levels generally declined, similar to BDNF supplementation. Combined BDNF + NT-3 treatment caused a mixed output. Neuron survival/neurite sprouting and neurite elongation equaled NT-3 supplementation, while the BDNF in the mix tunes trk/p75^NTR^ expression in the same way as BDNF alone. Interestingly, adding NT-3 to BDNF abolishes the neurite elongating effect of BDNF alone. ES largely impeded sprouting performance in a distinct tonotopic way with worst results in the basal turn. It is worth mentioning that ES enhanced outgrowth slightly in the apical turn with two distinct stimulation patterns (10 µA and 500 µA) and hence reversed the tonotopic gradient of BDNF treatment. The negative effects of ES on basal neurons led to or were caused by an upregulation of trk and p75^NTR^ receptor IR levels in the base. 

Supplementation with low amounts (1–25 ng/mL) of BDNF or NT-3 improved the morphology of SGNs and glial cells and reduced the presence of moribund SGNs (not counted). The best outgrowth performance for NT-3 with 200 ng/mL already impaired glia and neuron density. Increasing ES amplitudes deteriorated SGNs and (satellite) glia and weakened their mutual cohesion.

## 3. Discussion

In the present work, we investigated the effects of the two most promising NTs, BDNF and NT-3, combinations thereof, and simultaneous ES, on neurite regrowth, receptor expression, and morphology of SGN explants of P6–7 mice.

### 3.1. The Organotypic Explant Culture System

Neither the age and sex of the mice nor the body side where the explants were extracted from influenced the outgrowth significantly. However, explants from P6 mice exhibited slightly longer neurites, which may be explained by higher trk/p75^NTR^ receptor expression. Moreover, explants from female mice produced a few more neurites. Expression of estrogen receptors in female postnatal development [64] and a positive regulatory effect of sex steroids on BDNF expression and signaling [65] are known, but if this effect also applies to isolated SGNs in explant cultures is questionable. In an earlier study, however, we found that explants extracted from P7 mice performed better in total outgrowth than from younger mice [17]. Explants from the left body side performed slightly worse than from the contralateral side. This may be due to the prolonged time until explants from the left side were transferred to the culture medium postmortem, as an actual difference between the right and left ears seems to be unlikely. Whether these factors actually influence the outgrowth remains to be further investigated, as our results did not differ significantly. Without NT treatment, explants from the middle turn showed the most and longest neurites, followed by the apical and the basal turns. The highest SGN cell densities in the middle turn [66] may have resulted in most surviving neurons, and mutual trophic effects may have boosted outgrowth performance. In general, all measured values quantifying neurite outgrowth correlated largely. This indicates either that the analysis method was not able to separate different characteristics clearly enough or that intrinsic synergies exist. For example, explants that exhibit many outgrowing neurites often fasciculated into bundles, which may have resulted in mutual support and, therefore, longer neurites. The weight of the mice hardly altered outgrowth characteristics. However, the projected explant area positively affected all outgrowth measures. We tried to merge all variables into a nonlinear regression model to describe the system as a whole, but the results were contradictory. The model adapts the regression coefficients mostly to compensate for the unknown number of neurons within the explant body, leading to inconsistent outcomes.

Dissection of P6–7 mice proved to be challenging as ossification in the cochlea has already started at this age, especially at the basal turn. Bone fragments occasionally impeded the preparation of bone-free spiral ganglion explants, resulting in varying explant sizes and, consequently, in varying amounts of neurons within the explants. We assumed that this might be the most influential confounding variable on the outgrowth performance of an explant and tried several methods to compensate for it. The projected area of the explants or the immunofluorescent staining intensity of the explant body showed only weak correlations with the number of outgrowing neurites in the control group and therefore did not reduce variation. The projected area changed substantially (from a wedge shape to a more roundish shape) during culture as observed with live cell imaging (Appendix A) and hence may be less useful. Furthermore, we attempted to use NeuN immunolabeling and 2-photon microscopy to count neurons within the explant body, but immunolabeling was not consistent throughout the explant. Thick tissue, poor antibody penetration, and cell necrosis may contribute to this fact. A similar issue was encountered previously with explants from rats older than P5 [18]. However, this procedure and serial sectioning to count neurons turned out to be too labor-intensive for a medium-throughput approach. Preliminary results with immunolabeled transferrin as a live cell marker (data not shown) suggest that most regrown neurites are still viable after 96 h in culture. Live cell imaging also indicates that neurite formation continues well beyond this point in time (Appendix A). Considering the trk and p75^NTR^ receptor expression in mice, we observed a strong decline within the first postnatal days, stabilizing around P7. This suggests that mice younger than P6–P7 may not represent a mature-like developmental state for neurons and be less suitable for translational research. Furthermore, we observed a flat apex-to-base trkB gradient in P0–11 mice and a p75^NTR^ gradient in P0 mice.

Variation within the experimental groups was the most influential issue throughout this study. As already mentioned, we tried several approaches (normalizing for explant area or staining intensity, regression analysis) to minimize variance, but only direct controls reduced it reasonably. For the same reason, the initially planned fitting of dose-response curves failed. A considerable difference in outgrowth was evident among individual mice and especially litters. Litter size influences several metabolic parameters [67] and therefore may also be a factor for potential SGN outgrowth performance, which was not considered here. 

### 3.2. Effects of Neurotrophins

Both NTs greatly increased the number of neurites throughout all cochlear locations. However, BDNF was more effective than NT-3. We observed similar results previously in our lab with single-cell cultures of auditory neuroprogenitors [11]. This has also been confirmed in studies on explants [11,14,18,19] and in vivo [30]. Especially after BDNF treatment, explants from the basal turn exhibited a significantly stronger increase in neurite numbers than the ones from the other turns. Several studies [23,24,30,31,33] found increased SGN survival in the basal region and partly attributed this to the method of BDNF application and formation of a base-to-apex concentration gradient. Such a base-to-apex gradient was confirmed in in vivo studies with radiolabeled NT-3 [68,69]. Beyond this gradient, we suggest from our data that the basal turn SGNs may also be more susceptible to NT treatment, which may also contribute to the above-mentioned in vivo BDNF results. NT-3 supplementation was also most effective in the basal turn. BDNF [70], as well as NT-3 [71], are expressed in an apex > base gradient in postnatal animals. This indicates that the fewer NTs are naturally available, the more effective NTs are in improving neurite outgrowth. NT-3 tended to reduce trkB IR levels in the apical turn and trkC IR in the middle turn, whereas BDNF reduced trkB &C IR in the basal portion of the cochlea. Hence, NT-3 and BDNF showed trends for distinct tonotopic effects on trk receptors at the level of the SGN somata. In contrast to previous findings [11,19,35,72], we could not identify an additive or synergistic effect of both NTs applied simultaneously. We speculate that the ratio of the total amount of available NTs to the cultured biomass may play a role, as the NTs are partly internalized after binding to the respective receptor and then degraded or recycled [73]. We used a relatively large amount of 400 µL of culture medium per explant in our study, offering plenty of NTs for the explants. Continuous combined BDNF and NT-3 administration in high concentrations via a mini osmotic pump [32] did not improve SGN survival compared to individual NT administration. Therefore, one possible explanation for the synergistic effect found by others may be a rapid depletion of individual NTs that is compensated by the additional NT. Another one could be that we used rather high concentrations of BDNF and NT-3 in our combined groups and, therefore, may have already reached the ceiling of outgrowth responsiveness with a single NT. Such a ceiling effect was also discussed in a recent study about single vs. combined NT treatments on SGN preservation and responsiveness in deafened guinea pigs [35].

Median neurite lengths increased with both NTs as well as with the combined treatment significantly, although BDNF outperformed both other groups. Interestingly, when NT-3 was added, the trophic effect of BDNF treatment on neurite lengths was reduced to the level of only NT-3 supplementation. This novel finding suggests that NT-3 has an inhibiting function on neurite extension in this setting. BDNF and NT-3 exhibit relatively similar dissociation constants [74]. Therefore, NT-3 could partially block TrkB receptor binding sites and prevent binding of the neurite-elongating BDNF. Existing literature gives a heterogeneous picture of the effectiveness of NTs on neurite elongation: positive [12,18,20,75], neutral [14,76] and even negative [11] outcomes were reported. One study described a similar finding in dissociated P3 rat SGNs [10], with NT-3 administration having no effect on lengths while BDNF did so. The mentioned studies however used a wide range of setups and especially different length evaluation approaches. Overall, BDNF could act as a booster for neurite outgrowth after the loss of afferent targets, while higher concentrations of NT-3 in the healthy cochlea [71] promote survival of SGNs and suppress neurite elongation.

Measurements on mouse cochlea µCT images [77] revealed that the radial distance between the center of the SG to the estimated location of CI electrodes is approximately 200 µm, which is exceeded by 86% of all neurites of NT-treated explants. This makes both NTs suitable to promote neurite outgrowth, long enough to reach even hypothetical electrodes situated along the lateral wall in mice. This distance is much less than 2 mm, which would be approximately required in the human basal turn (own measurements from human cochlea sections [78]). However, SGNs are considerably larger in humans than in mice and hence may produce longer neurites. Moreover, neurite extension did not stop after 4 days.

Application of NTs resulted in a visual increase in SGN soma size and had a trophic effect on glia cells from 1 ng/mL up to 50 ng/mL. Several SGCs surrounded the neuron somata, while higher concentrations (200 ng/mL) resulted in smaller SGN bodies and diminished the number of associated SGCs. This data expose the huge NT concentration range that promoted neurite outgrowth as trk receptors may be activated even at picomolar NT concentrations [79]. However, centrally located SGNs and SGCs in explants showed more frequent signs of degeneration, suggesting that the supply of nutrients and NTs may be an essential factor. Moreover, the biochemical characteristics of BDNF prevent broader diffusion [80].

After nerve injury or complete axotomy, like in our model, trk and especially p75^NTR^ receptor expressions rose considerably compared to the respective in vivo levels. Still, exogenous BDNF administration suppressed this increase or lowered p75^NTR^ IR. TrkB & C IR intensities were even similar to the P7 in vivo situation. NT-3 application, however, did not show any effect on expression levels. In rats, after deafferentation following hair cell loss, expression of trkB was diminished while p75^NTR^ was elevated [81]. P75^NTR^ upregulation in Schwann cells and SGC after nerve injury is supposed to promote neurite regeneration [82] by increasing NT binding affinity to trk receptors boosting trk signaling [83] and facilitating glial migration and axonal growth [84]. Our data agree with these findings for trkB and depict that an eventual rise in binding affinity may explain the low trkB IR through receptor internalization. The response to this enhanced trkB signaling may explain the boost in neurite outgrowth and elongation. Saturating concentrations of BDNF, such as we used here, were recently shown to downregulate trkB in neurons that co-express trkB and trkC, while picomolar levels [79] did not show this effect. This enhanced receptor internalization could explain the decrease in receptor IR. Exogenous NT-3 has no effect on trk and p75^NTR^ IR levels and is less effective in promoting neurite outgrowth. As NT-3 was found to enhance glial migration and BDNF did not [85], elevated p75^NTR^ levels may reflect this increased migration of Schwann cells. Glial p75^NTR^ is also suggested to form a concentration gradient of NTs by binding of NTs to surface p75^NTR^. Elevated surface p75^NTR^ levels act as a positive signal for neurite outgrowth by chemo-attracting and guiding resprouting axons [83]. The up-regulation of p75^NTR^ in Schwann cells is maintained until axon-Schwann cell contact and remyelination [82] are reestablished by nerve regeneration. In our case, lower p75^NTR^ immunostaining after BDNF supplementation may also indicate that after a considerable amount of neurite re-sprouting and establishment of stable glia-nerve contacts, glial p75^NTR^ was no longer needed for axon guidance or Schwann cell migration. Hence, expression was downregulated even before any remyelination started.

### 3.3. Effects of Electrical Stimulation

For the ES experiments, we used direct controls in each culture plate to normalize the results, aiming for a reduction of variation. This was successful, as the range of results decreased compared to the NT experiments, but on the downside, more explants were needed in total. The ES samples were cultured with 25 ng/mL BDNF, which was the best-performing treatment group in the NT experiment. The best-performing treatment was chosen to evaluate if ES could be beneficial or should only be switched on after neurite outgrowth in a potential clinical setting. When comparing the unstimulated controls with the equivalent NT group (ES CTRL vs. BDNF 25 ng/mL) across the whole cochlea, a distinct reduction of outgrowth was evident in the ES controls. The only experimental difference was the exchange of the polystyrene lids to the electrode inset. An evaporation test revealed that 120 µL more of the medium evaporated than with a normal lid. This equals to 5 µL per well or ~1% more evaporation, which is not very likely to have altered the results to such an extent. Another reason could be an impaired gas exchange caused by the different lid, but we were not able to test this hypothesis. Figure 1A reveals that 25 ng/mL BDNF performed exceptionally well when compared to the neighboring concentrations (20 and 50 ng/mL). Hence, we speculate that the fewer samples (*n* = 24) in this group may have overestimated the true effects at this concentration and, therefore, may have outperformed the much larger ES CTRL^whole cochlea^ group (*n* = 201). Nevertheless, as the groups within both experimental sets were compared with their respective controls, this difference should not impair the conclusions of both experiments. 

In general, ES decreased the number of neurites as well as neurite elongation considerably during the neurite outgrowth phase. A small dose-amplitude-response relationship could be identified. The 1-min-on-2-min-off pattern resulted in the least number of neurite endings despite delivering less total charge than the continuous stimulation with 1000 µA or even the 500 µA treatments. A 3 min timespan between initiations of stimuli may have triggered a stronger onset response while a sustained stimulation may imply adaptation and hence a less pronounced response. This particular pattern may also have contributed since it resembles the spontaneous firing pattern of immature hair cells at P7 [61] and hence acted as an inhibiting cue for branching. Moreover, chronic depolarization may prevent the inactivation of voltage-gated channels [86], such as L-type calcium channels in SGNs [87] and alter activity-dependent neuronal gene expression [88]. The relatively long duration of ES in an in vitro system suggests that not only increasing amplitudes are more harmful but patterned ES can indeed influence gene expression and the health of SGNs more negatively than chronical ES [86]. What is harmful to completely de-afferentiated neurons in their neurite regrowth phase may not apply to adult neurons in the cochlear implant situation. Anyhow, etiologies behind CI are diverse, and data about neuronal decline in CI users is too sparse to draw conclusions about any long-term effects of stimulation patterns and neuronal survival in humans.

Our ES profiles showed tremendous effects on SGCs, which consequently lose their close association with their neurons. NTs and other growth factors produced by SGCs are the major contributors to the sprouting response [89], so a loss or detachment of these cells may explain the decreased outgrowth performance. Multiple voltage-gated ion channels were also found in auditory glial cells with distinct intracellular localizations [87], so the impact of ES on glial cells is obvious. Explants from the apical turn were most robust when stimulated electrically. The apex-to-base gradient of SGN maturation results in a less developed state of apical neurons at P6–7 [90] and could influence this robustness. Other intrinsic factors may account for this tonotopical difference. Several in vitro studies confirm that SGN survival [57] and neurite lengths [57,58] or fiber densities [59] decreased in a dose-dependent manner with increasing charge densities. The influence of ES in vivo is much more disputed, ranging from positive [30,43,44,45,46,47,48,49,50] to no effects [51,52,53,54,55,56]. Most studies however evaluated SGN densities and survival, as fibers are hardly accessible for quantification, while we focused on outgrowth measures. One in vivo study on cats counted peripheral fibers and detected that ES promotes their survival [46]. The study design differed from ours in a way that neonatal deafened kittens were treated for up to 10 weeks with BDNF, then drug administration terminated, and ES started and continued for several months. This treatment enhanced neuronal survival of somata and peripheral fibers and resulted in improved electrically evoked brain stem responses (EABR) compared to the BDNF-treated group without ES. Resprouted fibers were not counted or compared between the BDNF + ES and BDNF groups, but the authors speculate that resprouted fibers may have contributed to the improved EABR results. Our data also highlight a pronounced impact of ES on the tonotopical expression of trkB and p75^NTR^ receptors. ES with BDNF supplementation reduces trkB levels only in apical neurons but acts contrary in the basal turn. Neuronal activity may facilitate the internalization of trkB receptors as ES of hippocampal neurons was found to markedly enhance trkB internalization [91]. This reads as the neural activity evoked by ES with BDNF is higher in apical neurons and coincides with robust neurite outgrowth and lower trkB IR levels. As mild ES even increased neurite length and number with BDNF, it could be useful to switch on a BDNF eluting CI first in the apex with a localized bipolar stimulation of the most distal electrodes. Neurons in the basal turn should not be affected by such stimulation. Basal neurons with a more impeded outgrowth during ES + BDNF may have upregulated or less internalized trkB, so immunoreactivity for trkB was higher. Activating ES in the basal portion of the cochlea may be advantageous after the termination of NT treatment, as shown in vivo [46]. As our approach does not differentiate between surface, cytoplasmic or activated forms of trk receptors, we can only speculate about the complex regulation of trkB receptor internalization and trafficking with ES. P75^NTR^ IR levels showed similar trends in the basal turn with ES + BDNF, although IR is located in the glial cells of the explant. High p75^NTR^ levels in the basal turn with ES + BDNF may reflect elevated neuronal decline as glial cells upregulate p75^NTR^ after nerve injury [82]. Middle-turn neuron health status appears to be better, which resulted in the downregulation of neuronal trkC and glia p75^NTR^, while trkB levels were unaffected. This reflects the pronounced tonotopical difference of SGNs along the cochlea spiral. Overall, ES results are quite variable among in vivo and in vitro studies. In vitro studies lack the complex environment, such as vascular supply [92] or an ionic gradient [93], and differ in cellular maturation. On the other hand, in vivo work suffers from a low number of subjects and has a less repeatable methodology (drug distribution, electrode positions, inhomogeneous effects on supporting cell survival that may produce NTs). There is no report on the beneficial effects of ES in an in vitro study on SGNs [57,58,59,60]. We can largely confirm this negative effect of ES on regrowing SGNs, apart from apical SGNs with mild stimulation. 

We could not observe that the electric field acts as a cue for directing regrowing neurites, as detected in a previous study [94]. However, the methodology was different, as Li et al. tracked isolated neurites over time, and we measured the angles of the neurite endings regarding their presumed start-point from the explant. Our procedure is certainly less precise but enables the evaluation of a larger number of neurites. Nevertheless, we assume that we would have detected deviations from a random outgrowth orientation with our approach. Consequently, we cannot confirm our hypothesis that ES from a cochlear implant could be used to direct regrowing fibers toward the electrode contacts. Molecular cues such as an attracting extracellular matrix [95] or an NT concentration gradient of an eluting pump/CI [22] may be more useful for this task.

## 4. Material and Methods

### 4.1. Organotypic Explant Cultures

Organotypic explants of murine SGNs were prepared and cultured exactly as described previously [96]. Briefly, postnatal day (P) 6 and 7 mice of the C57BL/6N strain (Charles River, Sulzfeld, Germany) were used for the outgrowth experiments. The animals were bred at the animal facility in Innsbruck with unlimited access to food and water and a 12-h dark, 12-h light cycle. The animal studies conformed to the Austrian guidelines for the care and use of laboratory animals. The mice were decapitated rapidly, and both inner ears were removed. In ice-cold Hanks’ balanced salt solution, the bony capsule was removed, and the stria vascularis and the organ of Corti were torn from the spiral ganglion. The remaining spiral ganglion was separated into three half-turns. Each of the half-turns was then further freed from bony debris and the central nerve and then split into two equally sized explants, resulting in a total of six explants per ear (2 apical, 2 middle, 2 basal). Each explant was transferred to one well of a 24-well plate, already containing 400 µL of the culture medium, optionally supplemented with different concentrations of BDNF (450-02, PeproTech EC Ltd., London, UK), NT-3 (450-03, PeproTech) or combinations thereof. The culture plate was placed into an incubator at 37 °C and 5% CO_2_, and the explants were manually positioned into the center of the well. At least two explants per turn in three repetitions were cultured for each experimental condition. After 96 h, the culture medium was aspirated, and a collagen matrix was placed on top of each explant to protect the delicate neurites from mechanical damage due to further processing steps. The explants in the gelled matrix were fixed with 4% formaldehyde for 1 h at room temperature.

### 4.2. Electrical Stimulation

To assess the influence of ES on outgrowing neurites, a custom-made electrode inset for 24 well plates was designed and manufactured (Figure 11A–C). In total, 12 wells served for ES and could be stimulated simultaneously with individual ES patterns, while the other 12 wells were dedicated to unstimulated in-plate control explants. The inset consisted of a laser-cut stainless-steel base plate, electrode holders, electrodes, wiring, and a connector. The base plate (3 mm, 1.4404 2B cold rolled steel, Laserhub GmbH, Stuttgart, Germany) was designed to accommodate three electrode holders for each of the 12 wells. The electrode holders were made from acrylonitrile styrene acrylate tubes (007572014220 and 007572016420, Modellbau-Profi Niewöhner GmbH, Darmstadt, Germany). One long tube (outer-⌀ 4.0 mm, inner-⌀ 2.0 mm, length 20.9 mm) was inserted into a hole in the base plate, with 12.9 mm of the tube protruding beyond the lower plane of the base plate. The long tube was fixed with two wider short tubes (outer-⌀ 6.0 mm, inner-⌀ 4.0 mm, length 5 mm) that were slipped over and glued to both ends of the long tube. Gold-plated sockets (41.0010, Stäubli Electrical Connectors AG, Allschwil, Switzerland) were soldered to wires and glued into the long tubes. These wires were then soldered to a DA-15 connector. The actual electrodes were made from a platinum-iridium wire (90% platinum, 10% Iridium, ⌀ 1 mm, Surepure Chemetals Inc., Florham Park, NJ, USA) to mimic CI biomaterial composition [97]. The bigger reference electrode was made of 28 mm platinum-iridium wire. Both ends (~9.7 mm) were bent perpendicularly, and the middle part was bent to resemble an arc with a radius of ~8.6 mm. The stimulation electrode was straight and had a length of 11.5 mm and a flat tip. As the three electrode holders were arranged as an equilateral triangle around the center of the well, the stimulation electrode was at the center point of the arc of the reference electrode, theoretically resulting in a radial electrical field. To ensure that all electrodes were aligned to the bottom of the wells, the electrodes were pulled out a bit, and the inset was then put into a 24-well plate and pushed down until the base plate was flush with the well plate. This process was repeated whenever electrodes were misaligned. When the wells were filled with 400 µL of culture medium, only the actual electrodes were in contact with the medium, while the electrode holders were not submerged. Two electrode insets were manufactured and used in this study. The electrode insets were kept in 70% ethanol when not in use. Before culture, the electrode insets were placed in sterile well plates and kept in the incubator for 24 h to ensure that the electrodes were warmed up to 37 °C and that the ethanol could evaporate. All ES experiments were conducted with a culture medium supplemented by 25 ng/mL BDNF. This was the best-performing supplement without ES and was used to assess if ES can further improve elongation and direct growth of neurites or should rather not be used immediately after NT application. Preliminary experiments inspired us to use the best-performing concentration on the cost of a possible ceiling effect for the number of outgrowing fibers but with the highest sensitivity to detect enhanced elongation and directed growth. After preparation, the explants were allowed 24 h to settle and to attach to the well bottom. Then, the normal lid was removed, and the electrode inset was carefully placed into the well plate. The duration of the ES was 72 h, while the total culture duration remained unchanged at 96 h.

A fitting connector and a ribbon cable allowed the electrode inset to be connected from the inside of the incubator to an actual stimulator of a cochlear implant (PULSAR_CI_^100,^ MED-EL Elektromedizinische Geräte Gesellschaft m.b.H., Innsbruck, Austria). Two stimulators were used and driven by a MAX box (MED-EL, Figure 11D,E). The MAX box was programmed with special research software (RIB2.dll), which allowed a free configuration of the stimulation parameters. Five different biphasic stimulation patterns (Figure 11F) were created. The pulse duration was set to 30 µs, the interphase gap to 2.1 µs, and the frequency to 1000 Hz while only the amplitude varied (10, 100, 500, or 1000 µA) in these continuous stimulation patterns. An additional pattern with an amplitude of 1000 µA was made, but the stimulation was running for 1 min, followed by 2 min without stimulation. The explants were electrically stimulated for 72 h after a settling period of 24 h.

To test if the stimulation inset alters the evaporation of culture medium, two plates were filled with 400 µL of medium per well. One was incubated with a normal lid, and the other one with the stimulation inset. The weight difference of the plates before and after 72 h in the incubator was then measured. Additionally, temperatures in the center of six wells were measured using a logger (GL840, Graphtec Corporation, Tokyo, Japan) and type K thermocouple temperature sensors (Strasser G.m.b.H. & Co. KG, Innsbruck, Austria). After 30 min of temperature compensation, ES was switched on in three wells at 1000 µA for 1 h.

### 4.3. Immunostainings and Histology

The outgrown explants were further processed and analyzed, as explained in Schmidbauer et al. [96]. Antibodies used in this study are listed in Table 1. In brief, unspecific bindings were saturated, and the tissue was permeabilized, followed by the application of the primary antibody (beta-III-tubulin, Tuj1) and, after three washes, the secondary antibody (Alexa 546). Finally, after further washes, one drop of Vectashield Antifade Mounting Medium with DAPI (H-1200, Vectorlabs, Burlingame, CA, USA) was applied.

For quantifying the trk- and p75^NTR^ receptors, the following treatment groups were chosen: NT CTRL, 25 ng/mL BDNF, 200 ng/mL NT-3, 25 ng/mL BDNF + 200 ng/mL NT-3, and 25 ng/mL BDNF + 500 µA ES. Additionally, ROIs of outlined Rosenthal’s canal in 5 µm sections of freshly excised and fixed cochleae of P0, P7, and P11 mice served as reference. After whole-mount image acquisition with fluorescent markers (see Section 2.4.), some explants from these treatment groups were excised from the 24 well plates after adding the powerful gelling compound HistoGel (HG-4000-012, Thermo Fisher Scientific, Waltham, MA, USA). These explants and the freshly prepared cochleae were cryoembedded and cut at 5 µm as described by Coleman et al. [98]. Sections of explants from different treatment groups and of cochleae were gathered on the same slides to achieve a high level of comparability. Each of these panels consisted of the same set of treatment groups, and in total, three individual panels were made, resulting in three replications per group. The panels were then immunostained for trk and p75^NTR^ receptors with 3,3′-Diaminobenzidin (DAB) visualization by a fully automated system (Ventana Discovery, Roche Medical Systems Inc., Mannheim, Germany), using its DAB-MAP kit (760–124) and an EDTA buffer heat-induced antigen retrieval, as described by Luque et al. [99]. Semithin sections were made as described earlier [22]. Briefly, HistoGel confined explants were excised and post-fixed in 1% osmium tetroxide. Then specimens were dehydrated with ethanol and embedded in epoxy resin. 1 µm thick sections were stained with toluidine blue at 80 °C. 

### 4.4. Image Acquisition, Processing, and Evaluation

Images of the outgrown explants were acquired as already described [96]. In short, the DAPI and Tuj1 channel of the explants were imaged with a 40× air lens using an inverted fluorescence microscope with a motorized stage. The immunostaining technique focus was laid on neurites, so individual SGN somata could not be separated in our images, but even thin neurites were sufficiently stained. The images were stitched in Image Composite Editor (2.0.3, Microsoft Corporation, Redmond, WA, USA), resulting in 16-bit TIFF images containing a DAPI and a Tuj1 channel. These images were then further processed with ExplantAnalyzer [96], a fully automatic open-source script for MATLAB (2019a, The MathWorks, Natick, MA, USA). This script performs some basic image filtering steps, including adaptive thresholding, and converts the outgrowth morphology into a mathematical graph, which is a mathematical representation of a network. Numerous characteristics can be derived from this graph, such as the number of neurite endings or an estimation of the length of every neurite. 

The slides for the neurotrophic receptor quantification were imaged automatically by a TissueFaxPlus microscope system (TissueGnostics GmbH, Vienna, Austria) equipped with a PixeLink PL-B623 camera (Pixelink, Gloucester, ON, Canada) and a 40×/0.95 air lens. Semi-Quantification was performed with the dedicated image analysis software HistoQuest 7 from TissueGnostics. Regions of interest (ROI) were drawn around the explant body of in vitro samples or around Rosenthal’s canal in cochlear reference sections. Artefacts, such as dirt, tissue folds, etc., were removed by an exclusion area. The total area measurement mode considers only positive staining without detection of the cell type. The algorithm requires some training to recognize positive immunoreactivity. Hence, several shades of brown were selected as reference shades until areas with DAB staining were detected reliably. The mean intensity of all DAB-positive areas was then calculated for each tissue ROI. Intensity values are given as immunoreactivity (IR) intensity or IR, which is a dimensionless unit ranging from 0 (no staining) to 255 (saturation). 

The results were automatically arranged in MATLAB according to the treatment group. Variation of the outgrowth experiments was reduced by normalizing the results to the controls. For the NT-treated groups, only a global control existed, while the electrically stimulated groups were normalized to the unstimulated controls in the same plate of each experiment due to higher variability in ES experiments. The three turns were pooled to estimate the NT/ES response of the whole cochlea. As the three turns in each group were not equally present, the lowest number of specimens among the three turns was used, and the same number of specimens from the other turns were selected randomly to avoid bias. To evaluate the effect of the three main groups (BDNF, NT-3, and BDNF + NT-3) on neurite outgrowth across all concentrations, we pooled all concentrations of one main group randomly with equal shares. Controls of these equally pooled random samples are termed NT CTRL^whole cochlea^, ES CTRL^whole cochlea^ or NT/ES CTRL^apical, basal, middle^ when only certain tonotopic regions were equally pooled for statistical comparisons. Controls containing all the explants are labeled as NT CTRL or ES CTRL. 

### 4.5. Statistical Evaluation

Statistical evaluation was performed in Prism (9.3, GraphPad Software Inc., San Diego, CA, USA). As all data in this study were not normally distributed, non-parametric tests were conducted. Correlations were computed by a Spearman’s rank correlation analysis, and the influence of categorical variables was evaluated using an unpaired, two-tailed Mann–Whitney test. The results of the different NT concentrations were tested using a Kruskal–Wallis test with a Dunn’s multiple comparison post-hoc test, comparing every group with the untreated control group. *P*-values were corrected using the Bonferroni correction. As the electrically stimulated samples were normalized with the unstimulated samples in the same plate, grouped, and then compared with these control groups, a groupwise Mann–Whitney test, followed by a Holm–Šídák correction, was used. The mean intensities of the receptor quantification experiments were also tested by a Kruskal–Wallis test with a Dunn’s multiple comparison post-hoc test, but not each group was compared with every other group to avoid an unnecessary strong correction of the *p*-values. Therefore, the mean intensities of the ex vivo sections were tested amongst themselves, and the mean intensities of the NT-treated or electrically stimulated explant sections were compared with the mean intensities of the post-natal day 11 sections and the untreated control explant sections. *p*-values are indicated as follows: ∗ *p* < 0.05, ∗∗ *p* < 0.01, ∗∗∗ *p* < 0.001, ∗∗∗∗ *p* < 0.0001.

## 5. Conclusions and Implications for Cochlear Implantation

In this study, we found that BDNF alone could achieve maximal outgrowth, while NT-3 did not further contribute to this outcome. On the contrary, the addition of NT-3 nullified the elongating properties of BDNF. However, both NTs are known to alter the functional characteristics of SGNs [100] and must be utilized with caution. Ideally, an opposing gradient of both NTs should be established to maintain SGN properties while promoting outgrowth. It is more important to establish directed neurite growth towards the stimulation electrodes with little radial spread to increase specificity. Our ES profiles could not trigger directional growth. Other approaches include light [101], fibrous scaffolds [95], ultrasound [102], or biochemical gradients [103], but they are limited by the complex and rather inaccessible structure of the cochlea. Our CI-like ES patterns could not boost or guide neurite outgrowth but decreased the number and length of neurites. Hence, during the first days of neurite outgrowth, a CI should be switched on earliest when outgrowth goals have been achieved. Eventually, some mild stimulation in the apex may be beneficial during neurite regeneration. Afterward, ES could help to maintain the state, as shown in numerous in vivo studies. 

Exogenous BDNF was most potent in the basal region, and trkB expression was lowest there. Interestingly, trkC IR also decreased with BDNF in the basal turn, while our CI-like ES was most harmful in the basal turn with elevated trkB intensities. Neurite outgrowth in the basal neurons is most effective with BDNF but also most vulnerable to ES while sprouting. Therefore, a bioactive CI releasing NTs could work best with lower BDNF concentrations in the base and higher in the apex. If it proves that intrinsic factors make apical neurons most robust against ES, an ES pattern with low amplitudes could even be beneficial as trkB levels decrease and neurites extend further. Neurite endings were also slightly increased in the apical turn at 500 µA, and trkB IR was considerably reduced. Although we did not work with a more complex in vivo model and our SGNs are not fully matured, we regard results helpful for further research on NT-supplemented CI implantation. Resprouting SGNs close to final maturation shall provide us with the characteristic responses to NTs and ES in older neurons that enter into resprouting after implantation. Therefore, basic responses should be very similar and be better controlled with ex vivo systems. BDNF not only stimulates sprouting but preserves SGNs after a trauma such as explantation or CI implantation and outperforms NT-3 to close the man-machine gap in CIs.

In the present work, we shed more light on the outgrowth dynamics induced by BDNF and NT-3 and the effect of simultaneous ES along the tonotopic axis in a model close to the in vivo situation which still allows precise neurite quantification.

## Figures and Tables

**Figure 1 ijms-24-02013-f001:**
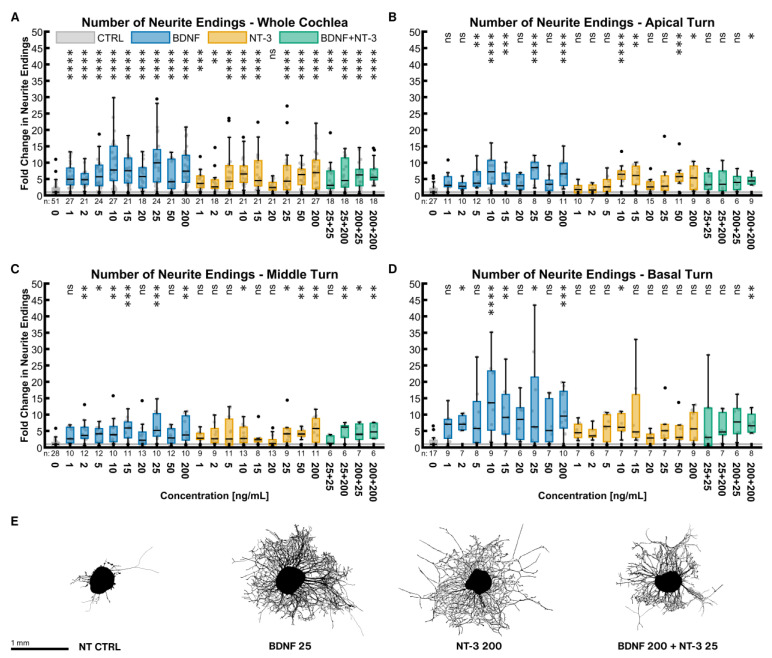
Number of neurite endings with neurotrophic supplementation. The diagrams (**A**–**D**) show the fold change of the number of neurite endings regarding the median of the NT control group with different neurotrophic supplementations. The groups in the first graph (**A**) contain an equal number of samples from three turns, hence representing the whole cochlea. Diagrams (**B**–**D**) depict the individual turns. (**A**–**D**) The boxes extend from the 25% to the 75% percentile. The black bar denotes the median. The whiskers delimit the 1.5-fold interquartile range. Samples within this range are marked as grey dots, and samples beyond this range are black. A grey line marks the median of the control group. The number of samples in each group is written right below the X-axis. The asterisks or ns (not significant) above the groups indicate the significance level of a Kruskal–Wallis test followed by a Dunn’s multi-comparison post-hoc test against the control group. (**E**) Representative explants at the median of the best-performing groups in (**A**).

**Figure 2 ijms-24-02013-f002:**
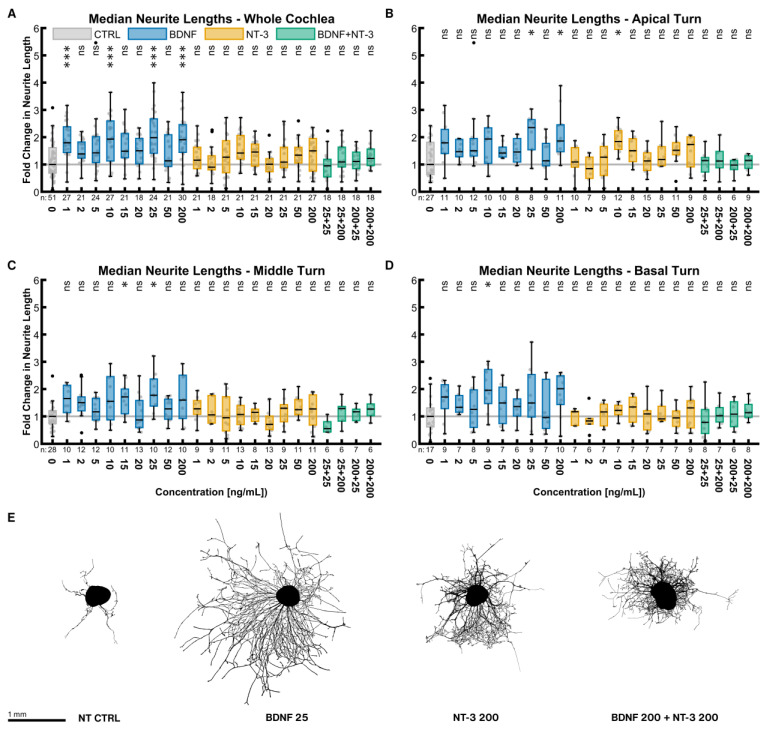
Median neurite lengths with neurotrophic supplementation. The diagrams (**A**–**D**) show the fold change of the median neurite length of every explant regarding the median of the control group with different neurotrophic supplementations. The groups in the first graph (**A**) contain an equal number of samples from three turns, hence representing the whole cochlea. Diagrams (**B**–**D**) depict the individual turns. (**A**–**D**) The boxes extend from the 25% to the 75% percentile. The black bar denotes the median. The whiskers delimit the 1.5-fold interquartile range. Samples within this range are marked as grey dots, and samples beyond this range are black. A grey line marks the median of the control group. The number of samples in each group is written right below the X-axis. The asterisks or ns (not significant) above the groups indicate the significance level of a Kruskal–Wallis test followed by a Dunn’s multi-comparison post-hoc test against the control group. (**E**) Representative explants at the median of the best-performing groups in (**A**).

**Figure 3 ijms-24-02013-f003:**
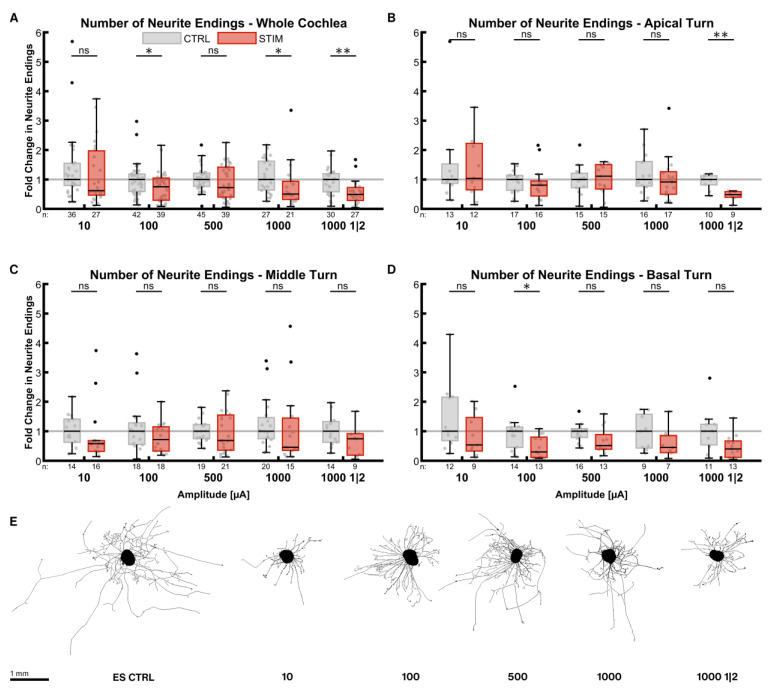
Number of neurite endings after electrical stimulation. The diagrams (**A**–**D**) show the fold change of the number of neurite endings regarding the median of the control group after ES with different amplitudes. All groups were supplemented with 25 ng/mL BDNF. The fold change was normalized to the electrically unstimulated controls on the same plate. Therefore, every electrically stimulated group has its own unstimulated control group. 1000 1|2 designates the 1 min on, 2 min off stimulation pattern with 1000 µA amplitude. The groups in the first graph (**A**) contain an equal number of samples from three turns, hence representing the whole cochlea. Diagrams (**B**–**D**) depict the individual turns. (**A**–**D**) The boxes extend from the 25% to the 75% percentile. The black bar denotes the median. The whiskers delimit the 1.5-fold interquartile range. Samples within this range are marked as grey dots, and samples beyond this range are black. A grey line marks the median of the control group. The number of samples in each group is written right below the X-axis. The asterisks or ns (not significant) above the groups indicate the significance level of a groupwise Mann–Whitney test, followed by a Holm–Šídák correction against the corresponding control group. (**E**) Representative explants at the median of the best-performing groups in (**A**).

**Figure 4 ijms-24-02013-f004:**
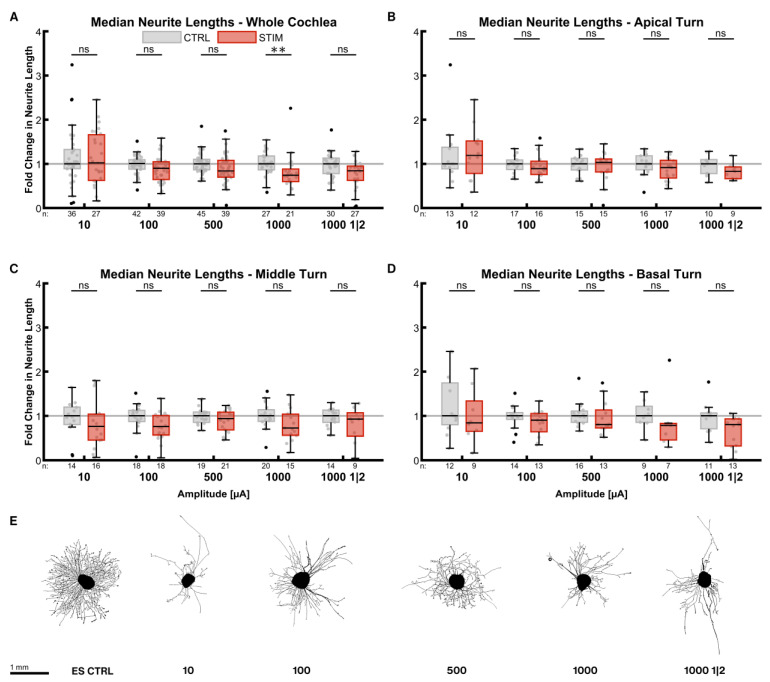
Median neurite lengths after electrical stimulation. The diagrams (**A**–**D**) show the fold change of the median neurite length regarding the median of the control group after ES with different amplitudes. All groups were supplemented with 25 ng/mL BDNF. The fold change was normalized to the electrically unstimulated controls on the same plate. Therefore, every electrically stimulated group has its own unstimulated control group. 1000 1|2 designates the 1 min on, 2 min off stimulation pattern with 1000 µA amplitude. The groups in the first graph (**A**) contain an equal number of samples from three turns, hence representing the whole cochlea. Diagrams (**B**–**D**) depict the individual turns. (**A**–**D**) The boxes extend from the 25% to the 75% percentile. The black bar denotes the median. The whiskers delimit the 1.5-fold interquartile range. Samples within this range are marked as grey dots, and samples beyond this range are black. A grey line marks the median of the control group. The number of samples in each group is written right below the X-axis. The asterisks or ns (not significant) above the groups indicate the significance level of a groupwise Mann–Whitney test, followed by a Holm–Šídák correction against the corresponding control group. (**E**) Representative explants at the median of the best-performing groups in (**A**).

**Figure 5 ijms-24-02013-f005:**
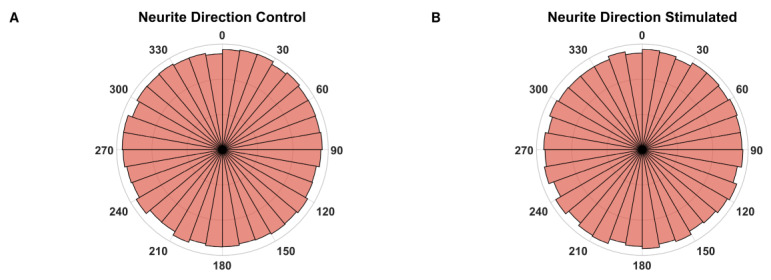
Growth direction angles. Polar histograms of the growth directions of every neurite from all electrically unstimulated (**A**, *n* = 112,850) and stimulated (**B**, *n* = 73,117) explants. The histograms are divided into 10° bins.

**Figure 6 ijms-24-02013-f006:**
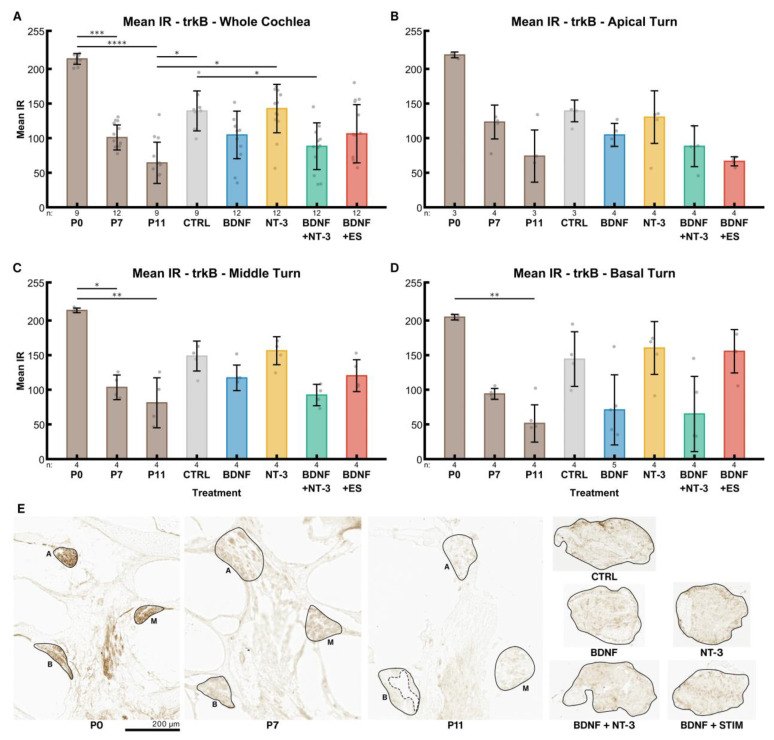
TrkB receptor expression. The diagrams (**A**–**D**) show the mean IRs of DAB-positive areas of histological sections in different age groups (P0–P11) and explants after different treatments (untreated (NT CTRL), 25 ng/mL BDNF, 200 ng/mL NT-3, 25 ng/mL BDNF + 200 ng/mL NT-3 and 25 ng/mL BDNF + 500 µA). The groups in the first graph (**A**) contain an equal number of samples from three turns, hence representing the whole cochlea. Diagrams (**B**–**D**) depict the individual turns. (**A**–**D**) The whiskers denote the standard deviation. Individual samples are marked as grey dots. The number of samples in each group is written right below the X-axis. The asterisks or ns (not significant) above the groups indicate the significance level of a Kruskal–Wallis test followed by a Dunn’s multi-comparison post-hoc test. The DAB intensities of P0–P11 pups were tested among themselves, and the treated explants were compared with the control and the P11 group. Non-significant differences between groups are not highlighted. (**E**) Representative sections closest to the median mean IR of the groups in (**A**). The solid black lines mark the regions of interest, and the dashed line represents an exclusion area. A, M and B denote apical, middle, and basal turns, respectively. The explant sections are all from the middle turn.

**Figure 7 ijms-24-02013-f007:**
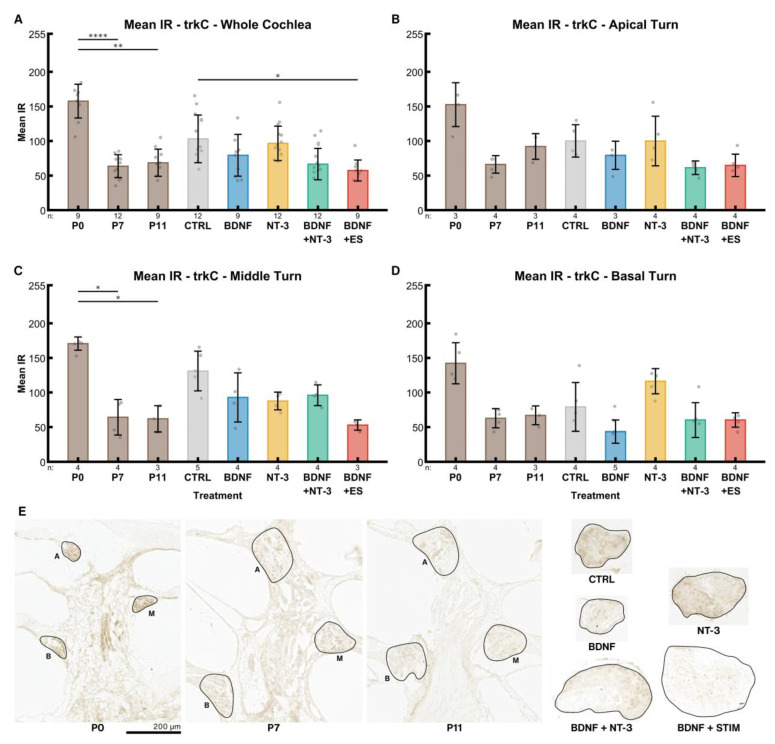
TrkC receptor expression. The diagrams (**A**–**D**) show the mean IRs of DAB-positive areas of histological sections in different age groups (P0–P11) and explant sections after different treatments (untreated (NT CTRL), 25 ng/mL BDNF, 200 ng/mL NT-3, 25 ng/mL BDNF + 200 ng/mL NT-3 and 25 ng/mL BDNF + 500 µA). The groups in the first graph (**A**) contain an equal number of samples from three turns, hence representing the whole cochlea. Diagrams (**B**–**D**) depict the individual turns. (**A**–**D**) The whiskers denote the standard deviation. Individual samples are marked as grey dots. The number of samples in each group is written right below the X-axis. The asterisks or ns (not significant) above the groups indicate the significance level of a Kruskal–Wallis test followed by a Dunn’s multi-comparison post-hoc test. The DAB intensities of P0–P11 pups were tested among themselves, and the treated explants were compared with the control and the P11 group. Non-significant differences between groups are not highlighted. (**E**) Representative sections closest to the median mean IR of the groups in (**A**). The solid black lines mark the regions of interest. A, M and B denote apical, middle, and basal turns, respectively. The explant sections are all from the middle turn.

**Figure 8 ijms-24-02013-f008:**
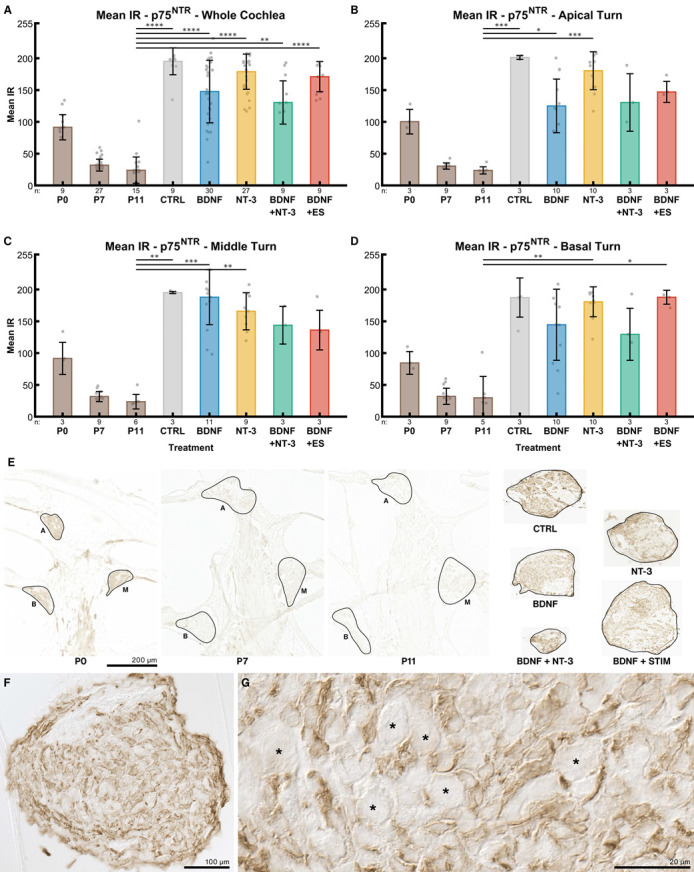
p75^NTR^ receptor expression. The diagrams (**A**–**D**) show the mean IRs of DAB-positive areas of histological sections at different age groups (P0–P11) and explant sections after different treatments (untreated (NT CTRL), 25 ng/mL BDNF, 200 ng/mL NT-3, 25 ng/mL BDNF + 200 ng/mL NT-3 and 25 ng/mL BDNF + 500 µA). The groups in the first graph (**A**) contain an equal number of samples from three turns, hence representing the whole cochlea. Diagrams (**B**–**D**) depict the individual turns. (**A**–**D**) The whiskers denote the standard deviation. Individual samples are marked as grey dots. The number of samples in each group is written right below the X-axis. The asterisks or ns (not significant) above the groups indicate the significance level of a Kruskal–Wallis test followed by a Dunn’s multi-comparison post-hoc test. The DAB intensities of P0–P11 pups were tested among themselves, and the treated explants were compared with the control and the P11 group. Non-significant differences are not shown. (**E**) Representative sections closest to the median mean IR of the groups in (**A**). The solid black lines mark the regions of interest. A, M and B denote apical, middle, and basal turns, respectively. The explant sections are all from the middle turn. (**F**) Overview and detail (**G**) of a 25 ng/mL BDNF-treated explant. SGNs are marked with an asterisk and do not exhibit any IR, while surrounding cells are considerably stained.

**Figure 9 ijms-24-02013-f009:**
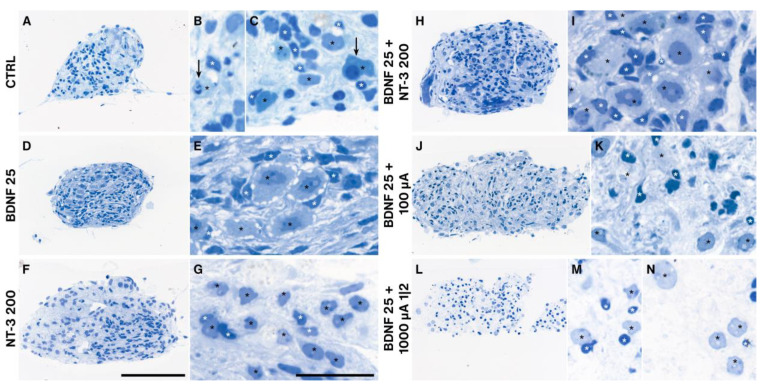
Semithin sections of BDNF and/or NT-3 supplemented, and electrical stimulated explant cultures. Treatments influenced the cell morphology of SGNs (black stars) and ensheathing SGCs (white stars). Control explants without neurotrophic supplementation (**A**–**C**) exposed many degenerating neurons (arrows) and several SGCs. Explants after 25 ng/mL BDNF treatment (**D**,**E**) resembled a normal morphology with many SGCs ensheathing SGNs with big nucleoli. 200 ng/mL NT-3 administration (**F**,**G**) resulted in big neurons but fewer SGCs associated with these neurons. Combined BDNF and NT-3 (**H**,**I**) application showed a similar appearance as with BDNF alone, while ES at 100 µA (**J**,**K**) and 1000 µA and a 1 min on, 2 min off pattern (**L**–**N**) lead to a marked decline of SGCs. Often SGNs located without their SGCs in the explant tissue (**L**,**N**). Scale bars: 100 µm (overview) and 10 µm (detail).

**Figure 10 ijms-24-02013-f010:**
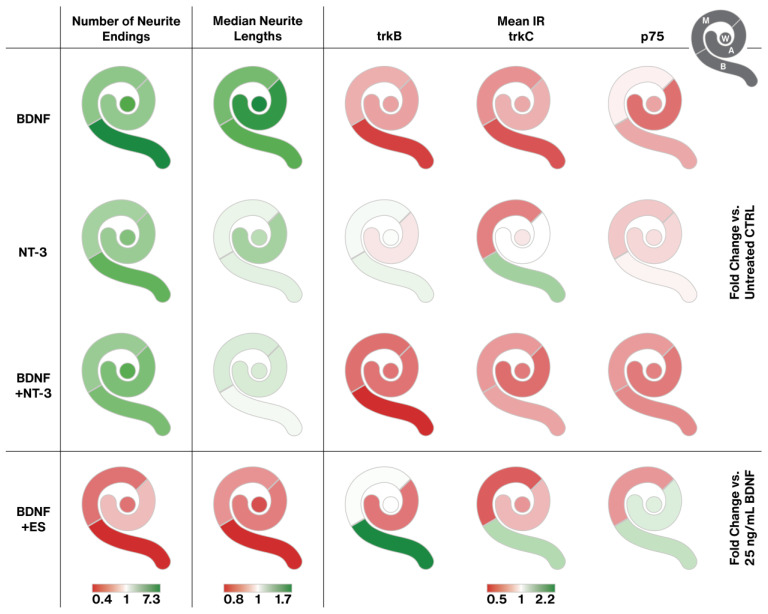
Graphical representation of the main results. Each cochlea pictogram is separated into four segments, representing apical (A), middle (M), and basal (B) explants, as well as the equally pooled, whole cochlea represented as a circle in the middle (W) (see pictogram in the upper right corner). The three gradients at the bottom determine the color range for the first, second, and last three columns, respectively. Each color visualizes the fold change of the median related to the median of the respective untreated (first three rows) or 25 ng/mL BDNF treated (last row) groups. To minimize the influence of variation, the first two columns contain all equally pooled concentrations or stimulation patterns of each agent (BDNF, NT-3, BDNF + NT-3, or BDNF + ES). For the mean IR of the receptor quantification, only distinct treatments were available (25 ng/mL BDNF, 200 ng/mL NT-3, 25 ng/mL BDNF + 200 ng/mL NT-3, 25 ng/mL BDNF + 500 µA ES).

**Figure 11 ijms-24-02013-f011:**
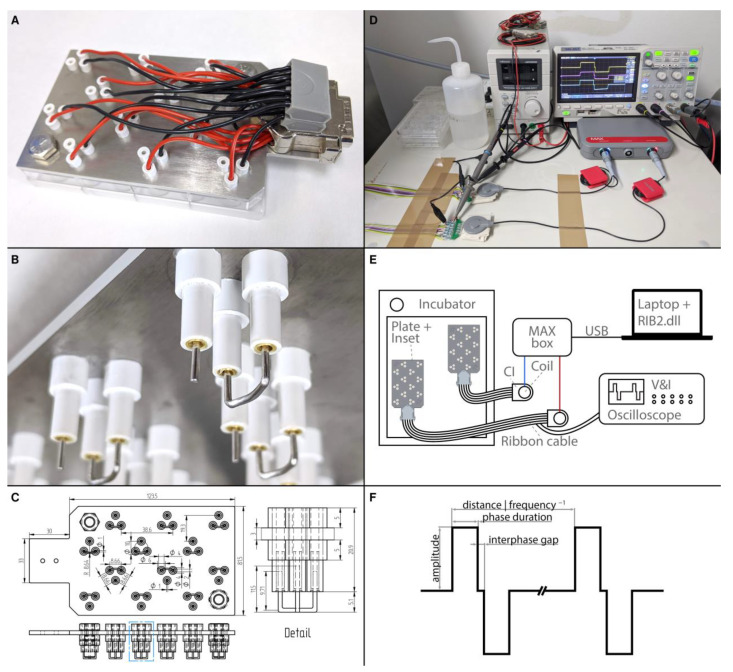
Electrical stimulation setup. (**A**) Photograph of the electrode inset in a 24-well plate. A set of electrodes is situated within every second well. (**B**) Detail of the lower part of the electrode inset. Three electrode holders fit into one well. The long and thin tubes are fixed by shorter and wider tubes above (not visible) and below the metal plate. In each holder sits a gold-plated socket that is soldered to a wire. The sockets hold the curved reference electrode as well as the stimulation electrode. (**C**) Multiview projections (top and side) and dimensions of the base plate, electrode holders, and platinum wires. The detail shows an enlargement of the features within the blue dotted box. Photograph (**D**) and schematic drawing (**E**) of the stimulation setup. Shown are two CI stimulators connected via coil transmitters to a MAX box. An oscilloscope monitors the voltage and current of the signal. Two ribbon cables lead into the incubator and are connected to the stimulation insets. (**F**) Schematic illustration of the stimulation signal and its main parameters.

**Table 1 ijms-24-02013-t001:** List of primary and secondary(marked with *) antibodies.

Antigen	Host	Company	Product	Dilution
Beta-III-tubulin/Tuj1	Rabbit	Abcam	ab52623	1:1000
Rabbit IgG + Alexa 546 *	Donkey	Thermo Fisher Scientific	A10040	1:1500
trkB	Goat	R&D Systems	AF1494	1:2000
trkC	Goat	R&D Systems	AF1404	1:1000
p75^NTR^	Goat	R&D Systems	AF1157	1:32,000
Goat IgG + Biotin *	Rabbit	Dako	E0466	1:400

## Data Availability

The explant evaluation tool ExplantAnalyzer, a MATLAB application, is freely accessible and can be downloaded here: https://github.com/DominikSchmidbauer/ExplantAnlayzer, accessed on 29 November 2022. The datasets produced and/or analyzed during the current study are available from the corresponding author upon reasonable request.

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
