# Peer review of "Closing the Gap between the Auditory Nerve and Cochlear Implant Electrodes: Which Neurotrophin Cocktail Performs Best for Axonal Outgrowth and Is Electrical Stimulation Beneficial?"

_ijms, 2023, doi:10.3390/ijms24032013_

Round 1
Reviewer 1 Report
General comment
The paper presents a well conducted in-vitro study on the effect of neurotrophins and electrical stimulation (ES) on neurite growth expressed in number and length. Both BDNF and NT-3 and the combination thereof are applied. The dose of the neurotrophins is varied over a wide range and also the ES is varied using 5 different strenghts/patterns. There is a complex amount of data, and I feel the authors could improve the paper by clarifying several of their arguments and by a clear summary of the data. See below specific comments.
Major comments
1) There is a mass of data orderly presented but at the end as reader one wonders how to summarize it. And importantly, how to answer the questions in the Title of the paper? A schematic of the main findings would strengthen the paper. The complexity of effects of the NTs and ES on number of neurites, neurite length, and receptors would then be more clear to the reader. For instance, the benefit of ES for apical neurons as mentioned in the Abstract, is not easily derived from the Results section.
2) The authors should explain why BDNF (in the best performing dose) was added for the ES experiments. The authors should further discuss the possible ceiling effect by adding an optimal dose of BDNF: there was hardly room for more neurite growth. In other words, possible outcomes would be rather negative than positive.
Minor comments
Abstract
p. 1 Line 29, and line 31: What is meant with high and low frequency neurons? Probably, basal and apical neurons are meant, and here in throughout the manuscript these terms should be used. The neurons are supposed to respond to electrical stimuli, not to acoustic tones.
p. 1 Line 31-32. The ES benefit should be specified here.
Introduction
p. 2, line 91-93. The study is supposed to shed more light on benefits of ES. However, as the authors have mentioned (line 77) there is controversy about the benefits of ES, with a wide variety of outcomes. I wonder how this study will somehow solve the controversy. I have not seen a discussion on the ES data of this paper in comparison to the literature.
Methods
p. 4, 5, line 145-170. A schematic drawing of the ES set-up would help. The images of Fig. 1A, B help, but a schematic along with the text would make it easier to understand.
p. 5, line 174. Remove “be” between “to” and “ensure”
p. 5, line 175-176. The authors should explain why BDNF (in the best performing dose) was added for the ES experiments.
p. 5, line 181. “cochlea implant” change to “cochlear implant”
p. 5, line 184. Remove “a” between “with” and “special”
p. 5, line 185-187. It may be added that these ES patterns involved continuous stimulation, as opposed to the fifth pattern.
p. 5, line 189-190. Was ES started at 24 h, and analyses performed on the samples immediately after offset of ES? This should be clarified. The wording of line 439 (section 3.3) can be used.
p. 5, line 208. Typo: first P7 should be P6.
p. 5, line 208. What of the mice served as reference? The Rosenthal’s canal?
p. 5, line 211. cochlea in “these explants and the cochlea”: should it not be “cochleas”
p. 6, line 239-240. Maybe add “reference” to “Rosenthal’s canal (see p. 5 line 208).
Results
p. 7, line 289. Why was BDNF added? See also comment above (p. 5, line 175).
p. 7, line 293. The two ages may be mentioned here for clarity.
p. 7, line 297 – 307. The P values should be mentioned both for significant and not-significant outcomes.
p. 7, line 297 – 307. Considering the large variance the precision of the median lengths should be reduced to 0 decimals, e.g., 340 µm and 326 µm.
p. 8, line 318. “Mice body weight” change to “Mouse body weight”
p. 8, line 318 – 321. The P values should be mentioned both for significant and not-significant outcomes.
p. 9, line 353-354. It may be stressed that the dose-response curves are quite noisy without a clear maximum response, therefore a best dose is hard to assess. For instance, the response for NT-3 does not considerably vary with dose from 5 to 200 ng/ml, apart from a minimum at 20 ng/ml (Figure 2A). I understand the authors’ approach to derive the best dose (find the maximum, irrespective of the remainder of the curve). Have the authors considered smoothing the curve before deriving a best dose? See also Discussion p. 25, line 820-822, where the authors discuss the exceptional outcome for BDNF at 25 ng/ml.
p. 10, line 396-398. First the results of BDNF are described with a P=0.0001, and then the NT-3 is “likewise” but then with p>0.9999. Could the authors clarify the P>0.9999? If not significant then one cannot claim “likewise performing best”.
p. 10, line 400. Again, P>0.9999. Is this an error, or is it supporting the statement of “only a minor effect”? With P>0.9999 I would state “no effect”.
p. 12, line 430. The reduction by 0.40 and 0.52 g does not tell much without mentioning the total weight. In other words, how much is the relative reduction? The percentages are mentioned in the Discussion, but they also should be mentioned here in the Results.
p. 12, line 431-436. The numbers may be mentioned to describe the decrease in neurite numbers. Also the statistical method should be mentioned with the P value.
p. 12, line 4567-458. The P value should be mentioned and the words after the comma may be deleted, since they are not significant.
p. 12, line 468. “bad” change to “poor”.
p. 13, line 488-490. The order of words should be changed, something like: neurite length decreases with increasing current level, resulting in a significantly shorter length at 1000 µA.
p. 16, line 538-547. The IR intensity is expressed with two decimals, which is too precise. Zero decimals would be sufficient considering values ranging from 64 to 214. Same for next sections. Also, what is the unit of the IR intensity?
p. 19, line 607-609. A description like this (tonotopical gradient of IR, location of IR) lacks in 3.4.1 and 3.4.2 about TrkB and TrkC.
Discussion
p. 22, line 664-666. Why are the differences with sex and age mentioned while they were not significant? Could the age effects be related to the decrease of Trk and p75NTR receptors with age (Figures 7-9)?
p. 22, line 674-675. The question is whether they are real left-right differences. The authors may explicitly state that.
p. 23, line 734-736. Do the authors have an explanation/speculation on the differences between their in-vitro findings and the in-vivo findings of Sugawara et al.?
p. 23, line 742-744. The reason for Staecker et al. (1996) not finding improved survival for the combined BDNF and NT-3 treatment seems to be a ceiling effect for individual NT treatment: there was no room for improvement since the NTs protect but do not regenerate neurons. The authors may want to check a recent paper of Vink et al. (Front. Mol. Neurosci., 2022) on the combined treatment vs individual treatment in conditions in which there was no ceiling effect.
p. 24, line 768-769. Could the authors provide a reference to support the 2 mm distance in the human cochlea?
p. 24, line 790-791. Is this based on literature or own findings?
p. 24, line 791-792. It would read easier when the authors mention that their data agree with previous statement (line 790-791).
p. 25, line 818. The numbers and percentages should be mentioned in Results as well.
p. 26, line 822-827. I agree with the reasoning. However, as raised above, did the authors consider smoothing the dose-response data?
p. 26, line 832-833. I would add to this that onset responses are larger than sustained responses as also demonstrated in auditory nerve physiology. Each 3 minutes a large onset response is evoked, which will be much stronger than the sustained responses in continuous stimulation. Compare to listening to a train of short tone pips vs one long tone.
p. 26, line 839-840. It may be explicitly added: “negatively influence”. How does this relate to cochlear implantation and the clinical situation in which CI users will be exposed to patterned ES? Can the authors comment on this?
p. 26, line 847-849. This sentence is not clear. Is it the less developed state of the basal turn and the apex matures earlier than the base?
p. 26, line 852-853. The authors may want to repeat the literature on this (like in the Introduction), for the reader who specifically reads the Discussion.
p. 26, line 855-856. It seems relevant to mention that the Leake et al. (2013) study also looked at BDNF in combination with ES. What was effect of ES without BDNF and that without BDNF?
p. 26, line 858. Here and at other places, change to “apical” for “low frequency” and “basal” for “high frequency”.
p. 26, line 861. “neural activity of ES” change to “neural activity evoked by ES”.
p. 26, line 861 – 868. The benefit of ES is discussed here, but not clearly. It should be explained better what the benefit is for the apical neurons, and at the same time, how does it outweigh disadvantages for basal neurons.
p. 27, line 872 – 873. It should be mentioned in which respect the current ES data contradict the in-vivo data, since the in-vivo data contradict each other in the literature.
p. 27, line 833. “cochlea implant” change to “cochlear implant”.
p. 27, line 915. What is meant here with “explanation”?
Figures
Suppl Figure 1. The diagonal with r=1 should be left empty as it represents trivial values and distracts from the actual data.
Figure 5, legend line 504. “number of neurite endings” change to “median neurite length”.
Figure 9, legend line 624. “comparisons” change to “differences”, and bold letter change to regular font.

Reviewer 2 Report
The article is scientifically sound, with interesting observations regarding the single and combined use of NTs and plenty of methodology details; for that matter, I suggest a clearer presentation of the methods used.
Other comments:
line 33: man-machine interface
line 71: and others: the different colors of the text (black and grey)
line 136: Figure 1 is not mentioned in the text
line141: two SI simulators - did you mean CI?
line 208: P0, P7 and P7- did you mean P11?
line 883: did you mean cochlear implant?
Author Response
The authors thank the reviewer for the time and effort to go through our manuscript and provide feedback. We are pleased that our work is appreciated and use the comments to improve the manuscript.
Comments and Suggestions for Authors
The article is scientifically sound, with interesting observations regarding the single and combined use of NTs and plenty of methodology details; for that matter, I suggest a clearer presentation of the methods used.
The reviewer is right. We inserted an introductory paragraph at the beginning of the material and methods section to provide an overview about the methods we used and for which purpose.
- Material and methods
In this study we investigate the acute effects of BDNF, NT-3, combinations thereof and ES on neurite regrowth of organotypic explants along the tonotopic axis in a medium throughput setup. Apical, middle and basal SGN explants were cultured for 4 days with different NT concentrations added. The best performing neurotrophin cocktail was further used for ES to identify whether ES can improve elongation and direct neurite outgrowth or should not be used immediately after NT administration. After fixation, neurites (number & length) as well as p75NTR /trk-receptor immunostainings were quantified. Plastic embedded sections exposed detailed morphology of each treatment group. Additionally, we shed more light on organotypic explant cultures of the spiral ganglion as a model for neuronal regeneration in the cochlea by analyzing correlations of categorical measures like sex, body side and age and numerical factors like explant area, branching points or body weight on neurite outgrowth.
We further incorporated comments from another reviewer to improve this section and provide detailed information for transparency of methods and reproducibility.
Other comments:
line 33: man-machine interface
The idea behind this grammatically incorrect use of the colon was to visualize the interface between the auditory system and a technical device in a figurative way. What may be usable in the headline of a grant application is maybe inappropriate in an abstract of a scientific manuscript, so we changed as the reviewer suggested.
line 71: and others: the different colors of the text (black and grey)
Corrected!
line 136: Figure 1 is not mentioned in the text
We added this in the first sentence in section 2.2. Electrical stimulation: To assess the influence of ES on outgrowing neurites, a custom-made electrode inset for 24 well plates was designed and manufactured (Figure 1A-C).
line141: two SI simulators - did you mean CI?
Corrected!
line 208: P0, P7 and P7- did you mean P11?
Yes, the reviewer is right. We corrected accordingly.
line 883: did you mean cochlear implant?
Yes, we corrected that.
